# PETAL: Physics Emulation Through Averaged Linearizations for Solving Inverse Problems

**Jihui Jin**
Electrical and Computer Engineering
Georgia Institute of Technology
Atlanta, GA 30332
jihui@gatech.edu

**Etienne Ollivier**
Mechanical Engineering
Georgia Institute of Technology
Atlanta, GA 30332

**Richard Touret**
Mechanical Engineering
Georgia Institute of Technology
Atlanta, GA 30332

**Matthew McKinley**
Mechanical Engineering
Georgia Institute of Technology
Atlanta, GA 30332

**Karim G. Sabra**
Mechanical Engineering
Georgia Institute of Technology
Atlanta, GA 30332

**Justin K. Romberg**
Electrical and Computer Engineering
Georgia Institute of Technology
Atlanta, GA 30332

## Abstract

Inverse problems describe the task of recovering an underlying signal of interest given observables. Typically, the observables are related via some non-linear forward model applied to the underlying unknown signal. Inverting the non-linear forward model can be computationally expensive, as it often involves computing and inverting a linearization at a series of estimates. Rather than inverting the physics-based model, we instead train a surrogate forward model (emulator) and leverage modern auto-grad libraries to solve for the input within a classical optimization framework. Current methods to train emulators are done in a black box supervised machine learning fashion and fail to take advantage of any existing knowledge of the forward model. In this article, we propose a simple learned weighted average model that embeds linearizations of the forward model around various reference points into the model itself, explicitly incorporating known physics. Grounding the learned model with physics based linearizations improves the forward modeling accuracy and provides richer physics based gradient information during the inversion process leading to more accurate signal recovery. We demonstrate the efficacy on an ocean acoustic tomography (OAT) example that aims to recover ocean sound speed profile (SSP) variations from acoustic observations (e.g. eigenray arrival times) within simulation of ocean dynamics in the Gulf of Mexico.

## 1 Introduction

Inverse problems arise in many scientific applications where the goal is to reconstruct some unknown signal, image or volume of interest from indirect observations. The forward process, or the mapping from the data to observations, is typically well-known usually through modeling the physical process. However, inverting the model is often ill-posed or even non-invertible. More formally, let us consider the task of recovering some signal $x$ from observations $y$ that are related by some potentially

37th Conference on Neural Information Processing Systems (NeurIPS 2023).

non-linear forward model $F$ via

$$\boldsymbol{y} = F(\boldsymbol{x}) + \boldsymbol{\eta}, \tag{1}$$

where $\boldsymbol{\eta}$ encapsulates noise or other perturbations. Our forward model $F$ represents a computational model of the underlying physics of the measurement process. Classical solutions involve modeling the forward process with extremely high accuracy and then attempting to invert a stabilized or linearized variant, which often requires heavy domain knowledge.

Another approach to handle the ill-posed nature of the inversion task is to cast the problem as an optimization task and incorporate regularization. A regularizer is a measure of how well the proposed solution fits some known, and often hand-crafted, prior. This term makes the inversion well-posed by biasing towards certain solutions. The inversion is typically solved for an in iterative fashion. This process can be potentially computationally expensive due to requiring many applications of the physics based forward model. Computing the adjoint to determine a descent direction increases the computational burden even more.

In this paper, we propose a novel architecture trained to emulate the physics-based forward model. The surrogate model provides a cheaper alternative for both the forward pass and descent direction calculation by leveraging existing auto-grad libraries[30], making iterative solvers feasible for recovering a solution. This work departs from previous works who also aim to emulate the forward model [4, 11, 33] by embedding low-fidelity forward models into the surrogate itself, thereby explicitly incorporating the physics instead of treating it as a black box. More concretely, we propose to use a set of linearizations of the forward model at a set of reference point and train a model to fuse and correct the outputs in an adaptive manner. We then use this trained model in a classical optimization framework to recover the signal given a set of observations.

Concretely, our paper makes the following contributions:

- We propose a novel architecture that learns to emulate the forward model. The model directly embeds physics via linearizations around a subset of reference points.
- We introduce a learned encoder/decoder scheme to the neural adjoint method to mitigate artifacts from directly optimizing in the input space.
- We demonstrate its efficacy for recovering solutions in a classical optimization framework on an ocean acoustic tomography (OAT) example that aims to recover ocean sound speed profile (SSP) variations from acoustic observations (e.g. eigenray arrival times) within a simulation of ocean dynamics in the Gulf of Mexico.

## 2 Related Works

### 2.1 Iterative Inversion

The task of directly inverting some potentially non-linear forward model $F$ is often non-trivial or mathematically impossible. Instead, a more stable alternative is to iteratively solve for $\boldsymbol{x}$ given some observations $\boldsymbol{y}$. Classically, this is done by formulating the reconstruction as solving a non-linear least squares problem that aims to minimize

$$\hat{x} = \arg\min_{x} \frac{1}{2} \|\boldsymbol{y} - F(\boldsymbol{x})\|^2 . \tag{2}$$

The Gauss-Newton method solves Equation (2) by iteratively computing the Jacobian $\boldsymbol{J}_F$ of $F$ at the current best estimate $\boldsymbol{x}^{(k)}$ and solving a set of linear equations to generate an update of the form

$$\boldsymbol{x}^{(k+1)} = \boldsymbol{x}^{(k)} + (\boldsymbol{J}_F^\top \boldsymbol{J}_F)^{-1} \boldsymbol{J}_F^\top (\boldsymbol{y} - F(\boldsymbol{x}^{(k)})). \tag{3}$$

The Levenberg-Marquardt algorithm [17, 23] presents a more robust alternative to Gauss-Newton by introducing a (non-negative) damping factor $\lambda$ leading to an update of the form

$$\boldsymbol{x}^{(k+1)} = \boldsymbol{x}^{(k)} + (\boldsymbol{J}_F^\top \boldsymbol{J}_F + \lambda\mathbf{I})^{-1} \boldsymbol{J}_F^\top (\boldsymbol{y} - F(\boldsymbol{x}^{(k)})). \tag{4}$$

Another approach to address the ill-posed nature of $F$ is to explicitly introduce a regularization function $R(\boldsymbol{x})$ as an additional penalty term to Equation (2). The regularizer $R$ addresses the ill-posed

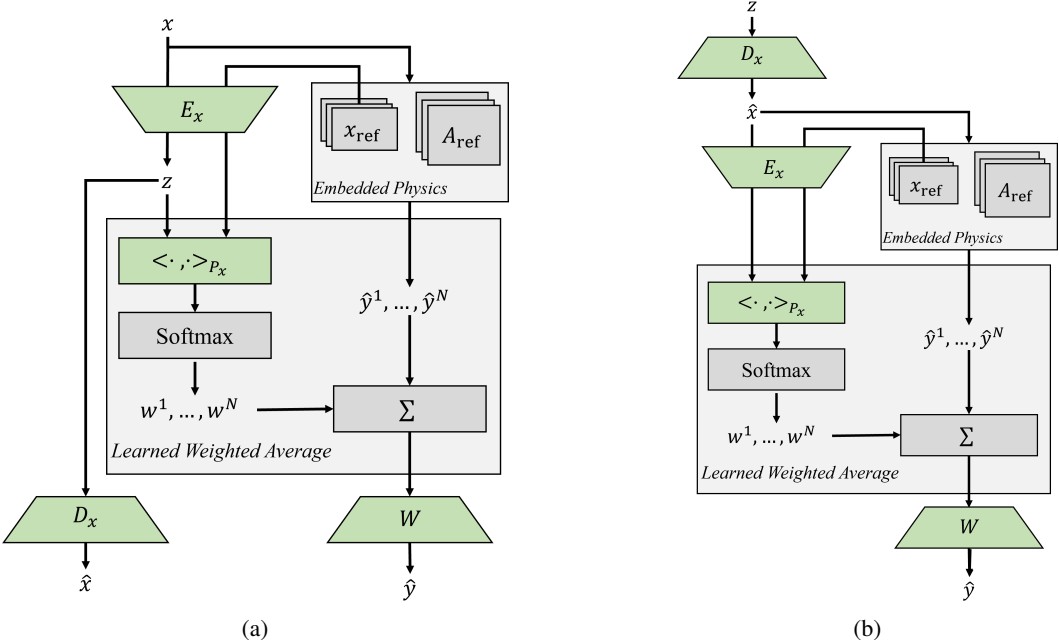

(a)                                                                     (b)

Figure 1: (a) Architecture of PETAL for training the forward model. The signal $\boldsymbol{x}$ is passed through the *Embedded Physics* module to generate low-fidelity physics-based predictions $\hat{\boldsymbol{y}}^1, ..., \hat{\boldsymbol{y}}^N$ that are then merged in an adaptive manner through attention mechanisms with the *Learned Weighted Average* module before being corrected with a final linear output layer $\boldsymbol{W}$. An encoding layer $\boldsymbol{E}_x$ and decoding layer $\boldsymbol{D}_x$ are simultaneously learned. (b) The architecture rearranged for inversion. The decoding layer is moved to the input to allow for inversion in the learned lower dimensional subspace $\boldsymbol{z} \in \boldsymbol{Z}$. Learned modules are denoted in green.

nature by stabilizing the inversion process and biasing the solution towards those with expected or desired structure. Some common examples include the $\ell_2$ norm to encourage smaller norm solutions (similar to Equation 4), $\ell_1$ to promote sparsity, and total variation (TV) to induce homogeneous regions separated by sharp boundaries. For differentiable $R$, the augmented optimization problem can be solved via steepest descent, computing updates of the form

$$\boldsymbol{x}^{(k+1)} = \boldsymbol{x}^{(k)} + \gamma \boldsymbol{J}_F^\top (\boldsymbol{y} - F(\boldsymbol{x}^{(k)})) - \lambda \nabla R(\boldsymbol{x}^{(k)}), \tag{5}$$

where $\gamma$ is a tuned step size, $\lambda$ is a non-negative factor controlling the strength of regularization, and $\boldsymbol{J}_F$ is the Jacobian of $F$ evaluated at $\boldsymbol{x}^{(k)}$.

However all these approaches can be undesirable in practice due to the need to re-compute the Jacobian (or a Jacobian-vector product) at each iteration. Computing a Jacobian-vector product to determine a descent direction often involves solving an auxiliary set of PDEs, which can be computationally expensive if many iterations are required to achieve an acceptable level of accuracy.

Ensemble Kalman filter (EnKF) offers a Bayesian iterative approach to estimating the underlying signal [8, 12]. Rather than estimating a single instance of the signal, EnKF recursively updates an estimated distribution of the state, assuming Gaussian probability. A sample mean and covariance is calculated using an ensemble of samples that are updated iteratively via Bayesian updates. However, such a method still ultimately relies on applying the physics based forward model to each member of the ensemble, which can be prohibitively expensive for complex non-linear forward operators.

## 2.2   Learned Inversion

An increasingly popular approach for solving inverse problems takes advantage of machine learning. Deep learning has achieved tremendous success in areas of natural language processing, computer vision and other tasks, in part due to the availability of large labelled datasets. Recent works attempt to tackle inverse problems using these data-driven methods [7, 13, 25, 29, 37, 38]. Unlike typical supervised learning tasks that attempt to learn a mapping purely from examples, deep learning for inverse problems can leverage our understanding of the physics in the forward model. One common approach to embed the physics is the so-called "Loop Unrolled" architecture heavily inspired by

existing iterative algorithms [1, 2, 9, 10, 35, 40, 41]. The learned model alternates between taking a gradient step computed directly from existing forward models and applying a learned regularizer step. However, such approaches have to apply the forward model multiple times for a single forward pass through the architecture, making the method infeasible for more complex non-linear simulators. Our approach bypasses this obstacle by using more computationally tractable approximations of the forward model.

An alternative approach to incorporating physics is termed "Physics Informed Neural Networks (PINN)" [16, 32]. These methods incorporate the governing partial differential equations directly in the loss, guiding a parameterized model towards physics-obeying solutions. However, an important distinction between PINNs and more generalized machine learning for inverse problems is that each model is trained for a *single instance* of a PDE solve given some boundary/initial conditions and must be re-trained for any new conditions or observables (in the case of inverse problems). Every training iteration involves a PDE misfit calculation, which can be expensive and ill-posed, making scaling to larger dimensions difficult. Unlike PINNs, the linearizations we use only need to be computed once before training rather than at each iteration of the training process.

Gaussian process regression (GPR) is another approach that leverages a set of references. GPR attempts to model some unknown function $f$ given data points by assuming a multivariate normal distribution prior and interpolating between the observed samples [39]. GPR is similar to the proposed model in that it does a distance comparison (usually through a pre-defined kernel function) with every reference point provided. However, it is designed to only take in input-output pairs and scales poorly with the number of examples, making it difficult to take advantage of large training sets. Our proposed model only needs to retain a subset of the training set for comparison purposes, but can still leverage large datasets for training a more accurate predictor.

## 2.3   Neural Adjoint

The neural adjoint method [33] was proposed to tackle inverse problems with more computationally intensive forward models. A parameterized model $G_\theta$ is trained to emulate the forward model [4, 11]. This is done in a supervised learning fashion, often with a simple mean-squared error loss. Not only does this provide a cheaper/quicker alternative to the physics-based forward model, if trained with existing auto-grad libraries such as PyTorch [30], this also allows for efficient computation of the gradient with respect to the input, bypassing the need to explicitly solve for the adjoint when calculating the gradient.

Once trained, the parameters are fixed and the model $G_\theta$ is substituted for $F$ in Equation (2) or other similar optimization frameworks. The auto-grad libraries are then used to efficiently compute a gradient with respect to the input $x$, making it possible to iteratively solve for the best estimate $\hat{x}$ that fits some observations $y$. Existing works primarily focused on lower dimensional examples (on the order of 5-15) where the test set was drawn from the same distribution as the training set [5, 22, 28, 31, 33], thus a simple "boundary-loss" regularizer was often sufficient to produce accurate results. Direct learned inversion methods are much faster than this iterative based method, but yield only a single estimate and are susceptible to overfitting to the training set. The neural adjoint method allows for exploration of the solution space with different initializations and the incorporation of various regularizers such as $\ell_1$, $\ell_2$, or even gradient based [21], to guide the optimization towards specific solutions. In addition, one can also restrict the optimization to some pre-determined basis that better represents the data while reducing the dimensionality [15].

Our proposed architecture extends the neural adjoint method in two notable ways. Existing methods are purely data-driven and thus treated like a black box [4, 33], while as our architecture explicitly incorporates knowledge of the physics-based forward model into the learned emulator in the form of linearized approximations. Tangent-linear models are also explored in [11], but on the surrogate in order to verify the accuracy of the learned model, while as the proposed architecture uses linearizations of the physics based forward model in the construction of the surrogate. In addition, we also propose to jointly learn a subspace to improve the accuracy of the optimization stage.

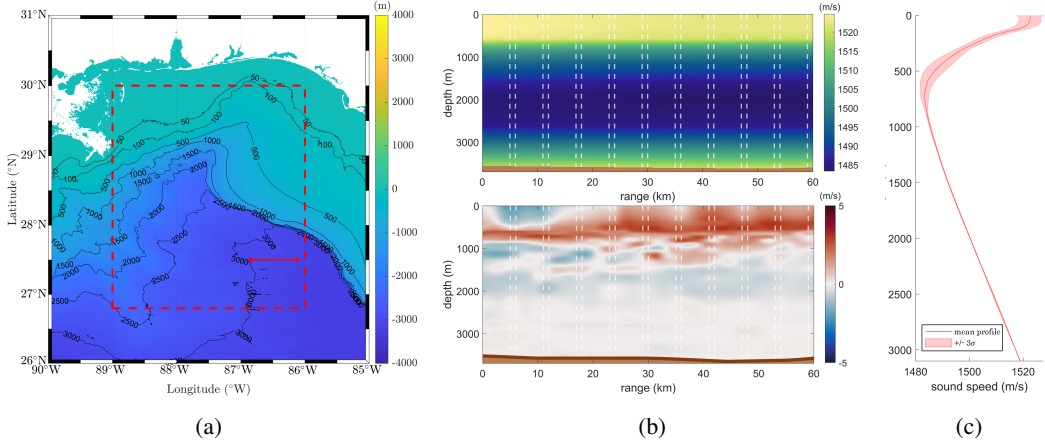

(a)                                    (b)                                    (c)

Figure 2: An overview of data from a month long simulation of the (a) Gulf of Mexico. (b) Example 2D slices of the sound speed profile (top) and its deviation from the mean (bottom). The 10 slices used for experiments are separated by white lines. (c) The average SSP as a function of depth. Note that most of the fluctuations occur near the surface.

## 3 Method

The neural adjoint (NA) method is typically decomposed into two stages: training an emulator of the forward model and then using it for inference. We motivate and describe the proposed architecture for the forward model in Section 3.1. Next, we formulate the optimization problem that uses the trained model for inference in Section 3.2. Finally, we propose an augmentation to the optimization formulation to incorporate a learned subspace in Section 3.3.

### 3.1 Embedding Physics in the Learned Forward Model

The neural adjoint (NA) method aims to learn an accurate emulator of the forward model to replace existing physics simulators. More formally, assume that we are given a forward model $F : \boldsymbol{x} \to \boldsymbol{y}$ that maps our input $\boldsymbol{x} \in \mathbb{R}^m$, where $m$ denotes the size of the discretization, to some observations $\boldsymbol{y} \in \mathbb{R}^n$, where $n$ denotes the total number of observations. In our computational ocean tomography examples in Section 4, $F(\boldsymbol{x})$ is computed using a ray tracer; in other applications, a PDE might have to be solved. We then train a neural net $G_\theta$, whose architecture is illustrated in Figure 1 and described in detail below, to approximate the mapping $F$.

We first motivate the design of the architecture by discussing the more classical approach of linearizing the forward model $F$. Given a reference point $\boldsymbol{x}_{\text{ref}}^i$, we perform a first order Taylor series expansion

$$
\begin{aligned}
\hat{\boldsymbol{y}} &\approx F(\boldsymbol{x}_{\text{ref}}^i) + J_F(\boldsymbol{x}_{\text{ref}}^i)^\top (\boldsymbol{x} - \boldsymbol{x}_{\text{ref}}^i) \\
&= \boldsymbol{y}_{\text{ref}}^i + \boldsymbol{A}_{\text{ref}}^i (\boldsymbol{x} - \boldsymbol{x}_{\text{ref}}^i).
\end{aligned}
\tag{6}
$$

This linearization approximates the forward model with varying levels of accuracy depending on the proximity of the input $\boldsymbol{x}$ to the reference point $\boldsymbol{x}_{\text{ref}}^i$. Rather than learning the mapping from $\boldsymbol{x}$ to $\boldsymbol{y}$ in a pure data-driven fashion, we propose to leverage a set of these linearizations that already perform the mapping. We mitigate the modeling inaccuracies by using an ensemble of reference models rather than a single linearization. Only a small subset of linearizations need to be performed once in order to construct the proposed emulator $G_\theta$, so the additional computational cost is minimal relative to attempting to invert the actual physics based forward model $F$ for arbitrary measurements $\boldsymbol{y}$.

The operation of the architecture acts as follows: Given some input $\boldsymbol{x}$, we first pass $\boldsymbol{x}$ through the embedded physics ensemble $\boldsymbol{A}_{\text{ref}}^1, ..., \boldsymbol{A}_{\text{ref}}^N$ to produce $N$ predicted observations $\hat{\boldsymbol{y}}^1, ..., \hat{\boldsymbol{y}}^N$ through the application of Equation (6). The predicted observations are then combined through a learned weighted average module to produce the final predicted observation.

A natural way to compute the weights $w^i$ is by calculating the dot product similarity between the input point $\boldsymbol{x}$ and the reference points $\boldsymbol{x}_{\text{ref}}^1, ..., \boldsymbol{x}_{\text{ref}}^N$ used to generate the linearizations; higher similarity implies that the linearization is a better approximation and thus the prediction is more "trustworthy". We follow an approach similar to attention-based models [36] by learning an embedding space to

perform the dot product and applying a softmax to normalize the weights. Thus for each $\hat{\boldsymbol{y}}^i$, we compute the corresponding weight $w^i$ as

$$w^i = \frac{\exp \langle \boldsymbol{x}, \boldsymbol{x}_{\text{ref}}^i \rangle_{\boldsymbol{P}_x}}{\sum_j \exp \langle \boldsymbol{x}, \boldsymbol{x}_{\text{ref}}^j \rangle_{\boldsymbol{P}_x}}, \tag{7}$$

where $\boldsymbol{P}_x$ is the learned projection for the dot product space. The predicted $\hat{\boldsymbol{y}}^i$ get summed together using the learned weights before being passed through a final linear layer $\boldsymbol{W}$. Thus the output of the proposed model is

$$G_\theta(\boldsymbol{x}) = \boldsymbol{W} \sum_i w^i (\boldsymbol{y}_{\text{ref}}^i + \boldsymbol{A}_{\text{ref}}^i (\boldsymbol{x} - \boldsymbol{x}_{\text{ref}}^i)), \tag{8}$$

where we distinguish components related to the embedded physics module with superscripts. The encoder-decoder layers $\boldsymbol{E}_x$ and $\boldsymbol{D}_x$ are treated as the identity mapping for this section. They are described in further detail in Section 3.3. Note that the learned weights $w^i$ depend on the input $\boldsymbol{x}$. To prevent saturation of the learned softmax distribution, we apply spectral normalization to all $\boldsymbol{x}$ projection layers ($\boldsymbol{E}_x$ and $\boldsymbol{P}_x$). The full architecture is outlined in Figure 1a. The model is trained using the mean squared error loss on a dataset of paired example $(\boldsymbol{x}_k, \boldsymbol{y}_k)$

$$\min_\theta \sum_k \|G_\theta(\boldsymbol{x}_k) - \boldsymbol{y}_k\|^2. \tag{9}$$

## 3.2 Performing Inference

In order to perform inference on a set of observations $\boldsymbol{y}$, we solve the following optimization problem that incorporates our trained network

$$\hat{\boldsymbol{x}} = \arg \min_{\boldsymbol{x}} \frac{1}{2} \|G_\theta(\boldsymbol{x}) - \boldsymbol{y}\|^2 + R(\boldsymbol{x}). \tag{10}$$

We solve this iteratively by fixing the weights of the network and computing the gradient with respect to its input $\boldsymbol{x}$. Note that this same optimization problem can be set up with the original forward model $F$, but computing a gradient is often non-trivial and computationally expensive. By training a forward model approximation, we can leverage existing auto-grad libraries [30] to efficiently compute the gradient. Having an accurate descent direction is critical for solving Equation 10. However, these black box models are only trained to match outputs, and thus performing gradient descent can lead to many undesireable local minima. Due to the construction of our emulator, a convex combination of the gradients from using the individual linear forward model approximations (slightly modulated by the learned weights) arises in the calculation, providing some physics-based descent directions which may help alleviate these issues (See Appendix for derivations).

Equation (10) can be solved with a variety of optimization algorithms to converge on some locally optimal $\boldsymbol{x}$. Since we are substituting the forward model with an approximation, we can account for any inaccuracies by introducing a tolerance level. Once the observation loss drops below a pre-determined level, the optimization terminates early.

Note that we incorporated a regularizer $R(\cdot)$ as an additional cost term. The regularizer encourages certain properties (e.g. smaller values) and helps guide the optimization towards particular solutions. In our experiments, we used $\ell_2$ as well as a Sobolev norm ($\ell_2$ norm performed on the discrete x and y gradient of our input $\boldsymbol{x}$).

Finally, it should be noted that the iterative nature of this method requires that we initialize our guess with some estimate. When optimizing from scratch, a reasonable candidate would be the average from the training set. Alternatively, we can leverage an estimated $\hat{\boldsymbol{x}}$ from other inverse methods by first initializing with that estimate and then refining it with the NA procedure. We note that there is an increase in computation time compared to directly learning the inverse due to the iterative nature of the algorithm, but the NA method offers certain trade-offs outlined above that might be more beneficial in practice than the single estimate provided by learned direct inverse methods.

## 3.3 Learning a Subspace for Reconstruction

Directly inverting on the input space of neural networks often generates artifacts in the final result or gets trapped in local minima due to the highly non-convex model. One way to address this issue is by

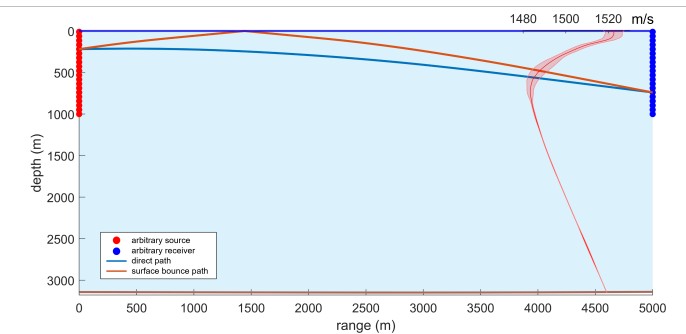

Figure 3: An example OAT forward model set up with 20 sources and 20 receivers. The direct and surface bounce paths are denoted in blue and red respectively. The average SSP influencing the arrival times at the receivers can be seen on the right.

Table 1: RMSE (m/s) of inversion with various initializations.

| Model | Avg Init | LFM Init | Tik Init |
|---|---|---|---|
| Tik | 0.760 | — | — |
| LFM | 0.608 | 0.625 | 0.602 |
| MLP | $0.391 \pm 0.005$ | $0.395 \pm 0.005$ | $0.393 \pm 0.005$ |
| PETAL (ours) | $\mathbf{0.364 \pm 0.018}$ | $\mathbf{0.357 \pm 0.014}$ | $\mathbf{0.357 \pm 0.014}$ |

optimizing in some lower dimensional subspace, such as one determined by principle component analysis (PCA) [15]. The PCA basis acts as a de-noiser, removing any artifacts due to the optimization and helps reduce the dimensionality, simplifying the optimization process.

Rather than pre-computing a basis, we instead propose to jointly learn a linear projection and reconstruction layer along with the forward model. The model as described in Subsection 3.1 can be augmented with a linear encoder $\boldsymbol{E_x}$ and decoder $\boldsymbol{D_x}$ layer. The encoder layer projects our input $\boldsymbol{x}$ onto a learned subspace, producing the latent code $\boldsymbol{z}$. This code is then passed through the decoder layer to reconstruct $\hat{\boldsymbol{x}}$. An additional input reconstruction loss in the form of a mean squared error loss between $\boldsymbol{x}$ and $\hat{\boldsymbol{x}}$ is included during training time, forcing the model to not only learn to approximate the forward model but also to learn a linear subspace of the input. We leave learning more complex non-linear encoder-decoder schemes for future work.

During inference, we then optimize in this subspace. More concretely, we rearrange the proposed architecture so that the $\boldsymbol{x}$ Decoder layer becomes the first input layer as shown in Figure 1b. The optimization variable $\boldsymbol{z}$ is passed through this layer to produce an estimated $\hat{\boldsymbol{x}}$, that is then passed through the rest of the model as described in Section 3.1. Our optimization framework thus becomes

$$\hat{z} = \arg\min_{\boldsymbol{z}} \frac{1}{2} \left\| G_\theta(\boldsymbol{D_x z}) - \boldsymbol{y} \right\|^2 + R(\boldsymbol{D_x z}), \tag{11}$$

where we recover our final estimate with $\hat{\boldsymbol{x}} = \boldsymbol{D_x} \hat{z}$.

## 4   Experimental Set Up

Although the method described in this article should be generalizable to any forward model, we demonstrate on an ocean acoustic tomography problem. Sound speed variations in the ocean are essential for accurate predictions of sound propagation and the various acoustic applications that rely on these predictions [6, 14, 24]. Typically, the ocean sound speed is estimated using empirical formulas based on the temperature, salinity and density. However, this would require a dense sampling both spatially and temporally throughout the whole volume of interest. Alternatively, the fundamental relationship between acoustic observations (e.g. arrival time measurements) can be leveraged to indirectly estimate the volumetric spatio-temporal variability of the ocean sound speed profiles (SSPs), bypassing the need to densely sample.

Ocean acoustic tomography (OAT) aims to reconstruct the SSP variations (with respect to a known reference environment) within an ocean slice given the changes in acoustic measurements from the

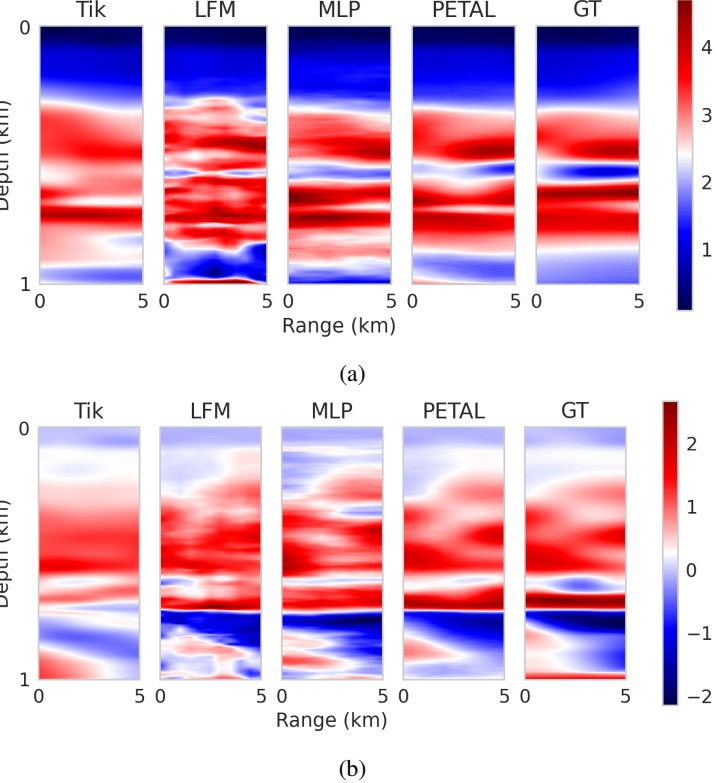

(a)

(b)

Figure 4: Example visualizations of predicted sound speed profiles for each method. Tik manages to recover general structure, but often gets the magnitude wrong. LFM introduces many artifacts during the optimization process due to the ill-posed matrix. MLP only captures coarse features but fails to capture subtler details like PETAL (ours).

propagating acoustic waves between multiple pairs of sources and receivers [26]. The "forward model" that computes arrival time measurements between source-receiver pairs given an SSP is fundamentally non-linear and would require solving the wave equation. However, SSP fluctuations are typically small compared to baseline values (typically $< 1\%$). Modeling assumptions can be made to simplify (e.g. ray-tracing methods). Classical OAT methods also further simplify by linearizing the relationship between SSP perturbations and the measured variations in arrival times of stable "paths" propagating between source and receiver pairs, providing a more numerically tractable solution (inverting a linearized model) [3, 26, 27, 34].

We perform our experiments on a high fidelity month long simulation of the Gulf of Mexico as seen in Figure 2 [19, 20]. We restrict our experiments to 10 2D range-dependent slices within the data cube (Figure 2b). The first 1000 time samples are used for training, the next 200 for validation and the remaining 239 for testing for each slice. This particular train/test/split in time was selected to mimic existing conditions (i.e. collect data for a set amount of time to train models and then deploy on future samples). Note that this creates a slightly more difficult problem due to the temporally changing dynamics in the test/validation set not present in the training set. The testset is further divided based on the variability within each slice. This is measured by the average deviation of the last 239 samples from the average of the first 1000.

The ray tracer BELLHOP simulates a forward model of sound waves being emitted from each of the sources. The receivers aim to detect peaks of the observed pressure wave, corresponding to the "arrival time" of the direct path and the surface bounce path of the sound wave. The simulation is an approximation of the wave equation. Rather than solving the full partial differential equations, the wave is modeled as an eigen-ray traversing through space, reducing it to an ordinary differential equation solve along the path of the ray between the source and receiver.

We experiment with two source-receiver configurations, a "dense" set up with 20 sources and receivers approximately every 50 m in depth and 5 km apart in range as shown in Figure 3. We also construct the more challenging "sparse" set up by sub-sampling in depth by a factor of 2. To counter the

differing bathymetry between the 10 slices, we restrict ourselves to the upper 1000 m portion of the ocean (where a majority of the SSP variation lies as shown in Figure 2c) and consider only direct or surface bounce arrival time paths. The 2D SSP is discretized into a range by depth grid of $11 \times 231$.

Thus, the linearized forward models (LFM) are of dimension $800 \times 2541$ (or $200 \times 2541$ in the sparse case), yielding an ill-posed matrix for inversion. We construct our proposed model with 10 reference SSPs: the last available SSP in the trainset (time 1000) for each of the 10 slices. Once trained, the weights are fixed and the ssp $x$ is solved for iteratively given some observations $y$. We perform the optimization in the learned subspace and with $\ell_2$ as well as Sobolev regularization. All optimized models are given a budget of 1000 gradient descent iterations with a learning rate of 50.

## 5    Results

We compare our proposed model against three baselines. First we compare ourselves against the pseudo-inverse performed in the PCA space and using Tikhonov Regularization as proposed in [15], hereby referred to as "Tik". We select the last available SSP in the train set (time 1000) for each respective slice as the reference point for linearization when evaluating on the test set. We perform PCA on the first 1000 SSPs for each respective slice to construct the basis. Next, we compare against using the linearized forward model (LFM) in an iterative method using the same regularization as the proposed model, but with the linearization around a single reference as the forward model. Similar to the Tik method, we linearize around the last available SSP in the training set for each respective slice. Finally, we compare ourselves with a simple multi-layer perceptron (MLP) trained to emulate the forward model. The MLP does not incorporate any physical information and is simply trained to map SSPs to arrival times in a black box manner. All iterative methods are initialized with: the average SSP, the Tik solution and the LFM solution.

The full results are summarized in Table 1. When provided no initialization (avg), the proposed method performs the best at $0.364 \pm 0.018$ m/s RMSE. MLP achieves the second best average performance at $0.391 \pm 0.005$ m/s RMSE, where the loss in performance is likely due to its inability to emulate the forward model as accurately as the proposed model. Despite using the same forward model set up, LFM (optimized with different regularization) is able to outperform Tik. We hypothesize that this is due to the basis computed by applying PCA to the training set failing to generalize to the dynamics of the test set. However, when allowed to optimize further (LFM init), degradation occurs due to the inaccuracy of the forward model approximation, increasing from 0.608 to 0.625 m/s error.

MLP also fails to refine the solution for both the LFM as well as the Tik initialization, suggesting that the learned non-linear forward model gets caught in local minimas despite having better initialization. Our proposed model achieved better results when initialized with the more accurate estimates, dropping the RMSE to 0.357 for both LFM and Tik initializations.

Visualizations of the recovered SSPs can be seen in Figure 4. All models are able to recover the background levels and other coarse features. Tik is able to recover some finer details, but often fails to achieve the correct magnitudes. LFM introduces many artifacts due to the ill-posed forward matrix, an issue mitigated by using a subspace in Tik and PETAL. MLP is able to capture some of the finer details, but also suffers from artifacts introduced during the non-convex optimization.

Table 2 breaks down the accuracy into further sub categories. Note that the level of variability does not always correlate with the performance of each method. For examples, PETAL performs the worst on the dense case with low variability with average initialization at 0.365 m/s RMSE. However, when provided with the Tik initialization (that performed better on the low variability scenario), it is able to achieve the best RMSE of 0.339 m/s. The sparse source/receiver scenario shows a degradation in performance across all methods as expected given the less observations available to each model. However, trends are still upheld with learned surrogates MLP and PETAL outperforming classical linearization based solutions and PETAL performing the best overall.

## 6    Ablations

In this section, we explore which design decisions contribute to the overall success of PETAL. More specifically, we try to determine whether the learned weighted averaging, learned transformation of the predicted arrival times or the learned encoding/decoding of the SSP are all necessary components.

Table 2: RMSE (m/s) of inversion across different scenarios.

| | Model | Low Variability | | Med Variability | | High Variability | |
|---|---|---|---|---|---|---|---|
| | | Avg | Tik | Avg | Tik | Avg | Tik |
| Sparse S/R Pairs ($10 \times 10$) | Tik | 0.724 | — | 0.967 | — | 0.926 | — |
| | LFM | 0.692 | 0.670 | 0.674 | 0.677 | 0.727 | 0.728 |
| | MLP | 0.465 | 0.456 | 0.477 | 0.494 | 0.463 | 0.466 |
| | PETAL (Ours) | **0.450** | **0.444** | **0.423** | **0.431** | **0.416** | **0.420** |
| Dense S/R Pairs ($20 \times 20$) | Tik | 0.647 | — | 0.773 | — | 0.881 | — |
| | LFM | 0.620 | 0.597 | 0.584 | 0.580 | 0.617 | 0.630 |
| | MLP | 0.384 | 0.378 | 0.406 | 0.409 | 0.424 | 0.428 |
| | PETAL (Ours) | **0.365** | **0.339** | **0.360** | **0.346** | **0.361** | **0.374** |

Table 3: Ablation study of PETAL

| Model | Weighted Avg | AT Transform | SSP Subspace | RMSE (m/s) |
|---|---|---|---|---|
| PETAL (ours) | ✓ | ✓ | ✓ | **0.343** |
| A-LFM | ✗ | ✗ | ✗ | 0.585 |
| WA-LFM | ✓ | ✗ | ✗ | 0.577 |
| WA-LFM + Dec | ✓ | ✗ | ✓ | 0.508 |
| WAN | ✓ | ✓ | ✗ | 0.372 |

The baseline will be the proposed model which incorporates all three design choices. We compare this against (1) the average of all reference linearizations (A-LFM), (2) weighted average of linearizations (WA-LFM), (3) weighted average combined optimized in the learned subspace (WA-LFM + Dec), and (4) a learned weighted average network (WAN). A full summary as well as the average RMSE on a test set can be found in Table 3.

This study shows that each component is essential in the success of the proposed model. A-LFM performs the worst overall, though still noticeably better than the single LFM presented in Table 1, suggesting that incorporating more references is crucial for representing the varied dynamics present in the test set. Simply adjusting the weights of the average (WA-LFM) as learned by a fully trained PETAL model already leads to an improvement in performance, dropping the RMSE from 0.585 to 0.577. Incorporating the learned SSP subspace improves even further, dropping the RMSE to 0.508. Learning an AT transform allows the surrogate model to better approximate the true model, leading to a more dramatic improvement in RMSE at 0.372 for WAN. And finally, incorporating all three components leads to the best performance overall at an RMSE of 0.343

# 7 Conclusions

In this study, we propose a novel architecture to embed physics in learned surrogate models by incorporating linearizations around a set of reference points. We also include an encoder-decoder structure to learn a subspace to optimize in when solving inverse problems, mitigating issues arising from the non-convex optimization. We demonstrate the efficacy of our approach on an Ocean Acoustic Tomography example, outperforming classical methods as well as learned forward models that do not incorporate any known physics. We validate the necessity of each component in an ablation study, confirming that each contributes to the success of the proposed model.

## Acknowledgments and Disclosure of Funding

This work was sponsored by the Ocean Acoustics program at the Office of Naval Research (Code 322) and the Task Force Ocean. We would also like to thank Dr. Guangpeng Liu and Dr. Annalisa Bracco (EAS, GaTech) for sharing Gulf of Mexico simulations of the SSPs.

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

Table 4: Architecture details of final proposed PETAL model.

| Layer | Input Dim | Output Dim | Spectral Norm? |
|---|---|---|---|
| $x$ Encoder | 2541 | 1000 | ✓ |
| $P_x$ | 1000 | 1000 | ✓ |
| $W$ | 800 | 800 | ✗ |
| $x$ Decoder | 1000 | 2541 | ✓ |

## A    Training the Forward Model

All experiments were performed on a GeForce RTX 2080 Super.

### A.1    Data Preparation

We normalize our data using the training set "pixel-wise" average and standard deviation for training only.

### A.2    PETAL

The proposed PETAL model only uses linear layers throughout. However, it is able to learn a complex non-linear model due to the attention-inspired mechanism. The exact details of each linear sub-component can be found in Table 4. The embedded physics layer takes in an input of dimension $BS \times N_x$, where $BS$ is the batch size and $N_x = 2541$ the size of the vectorized SSP, and outputs a tensor of dimension $BS \times N_{\text{ref}} \times N_y$ where $N_{\text{ref}}$ refers to the number of linearized reference models and $N_y = 800$ is the number of observations.

The weighted average layer inspired by attention layers takes in a query of dimension $BS \times 1 \times H_x$, where $H_x$ is the hidden dimension for the latent space of input SSPs, a key of $BS \times N_{\text{ref}} \times H_x$ and a value of $BS \times N_{\text{ref}} \times N_y$. It outputs a predicted arrival time vector $\hat{y}$ of dimension $BS \times N_y$.

The model was trained using ADAMW with a learning rate of 1e-5 for 500 epochs. The learning rate was dropped by a factor of 0.2 at epoch 300.

The model was trained to minimize the MSE of the arrival time prediction as well as a MSE on the SSP reconstruction. The selected model achieved training RMSE of 4.88e-2 and validation RMSE of 1.02e-1. It achievse an unnormalized (removing training normalization) AT RMSE of 4.98e-4 and SSP RMSE of 5.37e-2 on the validation set.

### A.3    MLP

We experimented with both encoder-decoder like structures as well as models without the bottleneck layers. The final best performing model had 4 hidden layers of dim 1500 with leaky ReLU non-linearities. It achieved a training RMSE of 3.62e-2 and validation RMSE of 1.22e-1. The unnormalized AT validation RMSE was 6.08e-4 (higher than PETAL). The model was trained using Adam for 250 epochs with a learning rate of 1e-5.

## B    Optimization Framework

The neural adjoint method is an iterative method to recover an SSP $x$ given some observations $y$. All models are optimized using Pytorch's Stochastic Gradient Descent with a learning rate of 50 for 1000 epochs.

We use two forms of regularization: an $\ell_2$ penalty on $x$ with a scale of 1e-7 and a Sobolev penalty ($\ell_2$ on the discrete x and y gradient) with a scale of 1e-4.

The optimization is performed in batches. We set an early cutoff rate of 1e-2 such that for any sample, if the forward model observation loss drops below this value, we cut off the gradient to that sample. This value is lower than the final (normalized) mse AT loss of any of the models, so the assumption is that any further optimization beyond this point will just overfit to the model.

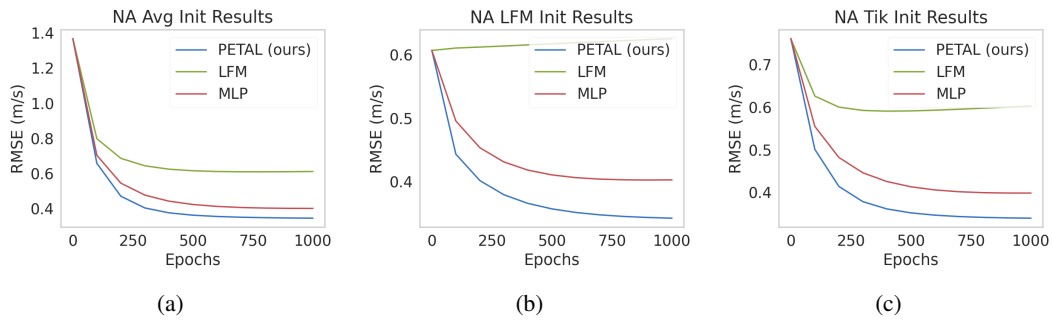

(a)                                    (b)                                    (c)

Figure 5: RMSE of different models vs number of epochs optimized given (a) average, (b) LFM, and (c) Tik initializations.

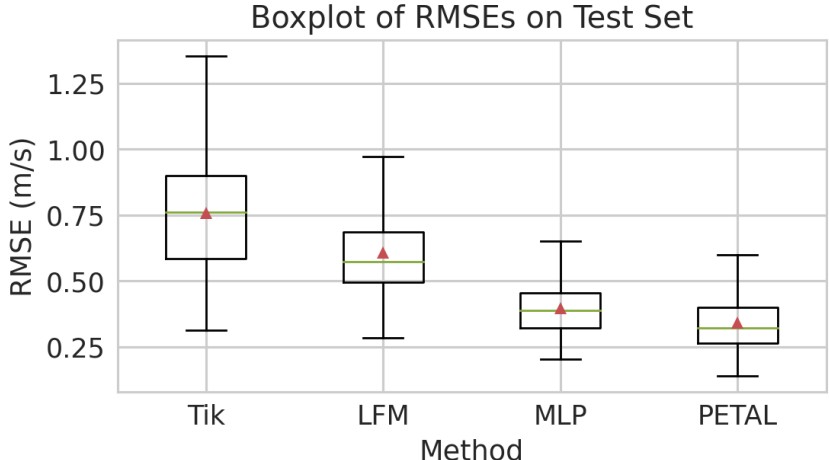

Figure 6: Boxplot of RMSE of different models.

## B.1   Results

The results of NA given different initializations for each of the forward models can be seen in Figure 5. Although further iterations might yield higher performance, the overall RMSE already begins to plateau around 1000 epochs. For some models (particularly LFM), the performance already starts to degrade. The distribution of errors after 1000 iterations can be found in Figure 6.

## B.2   Robustness to Unseen Slices

In this section we explore the robustness of surrogate models to unseen slices. We perform this experiment by training the surrogate models on only slices 1-9 (with the same train/val/test split) and then evaluating on the entirety of slice 10. The performance can be seen in Figure 7 and Table 5. We refer to the subsets of slice 10 as "Train","Val", and "Test" for convenience, referring to the temporal split of the data, but no samples from slice 10 were available during train time. We select the linearization around the last available SSP in the times corresponding to the train set for LFM.

Both trained surrogate models greatly outperform LFM in the subset of the slice overlapping in time with the trainset, suggesting that there are some shared dynamics across space that can be learned. Notably, most models begin to degrade in the times corresponding to the validation and test set, highlighting the difficulty in capturing dynamic shifts over time. However, the learned models still remained more robust to this shift and the performance only degraded slightly compared to when trained with all slices dropping from 0.33715 (when evaluated only on slice 10) to 0.33736 for our proposed model PETAL.

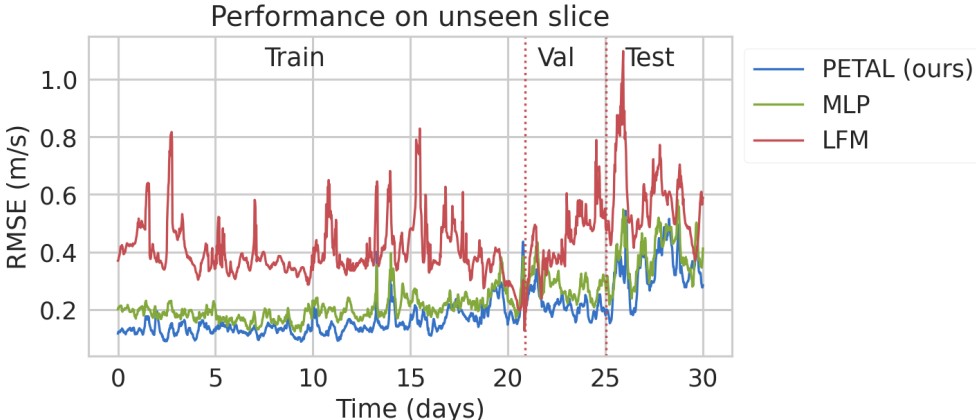

Figure 7: Performance of models on unseen slice. Both trained forward models perform well on the subset of the slice overlapping in time with the trainset, suggesting that dynamics are shared throughout the region.

Table 5: Average RMSE (m/s) of inversion on unseen slice.

| Model | Train | Val | Test |
|---|---|---|---|
| LFM | 0.405 | 0.447 | 0.583 |
| MLP | 0.196 | 0.288 | 0.402 |
| PETAL (ours) | **0.149** | **0.217** | **0.337** |

## C   Gradient of PETAL

Define a (simplification) of the PETAL model as

$$
\begin{aligned}
\hat{\boldsymbol{y}} &= \boldsymbol{W}(\sum_i w^i \hat{\boldsymbol{y}}^i) \\
&= \sum_i w^i \boldsymbol{W}(\boldsymbol{A}_{\text{ref}}^i \boldsymbol{x} + \boldsymbol{b}^i).
\end{aligned}
\tag{12}
$$

Note that by construction, the weights $w^i$ sum up to 1. If we include this in a simple MSE loss we get

$$
L = \frac{1}{2} \|\hat{\boldsymbol{y}} - \boldsymbol{y}\|^2.
\tag{13}
$$

Computing a gradient w.r.t. $\boldsymbol{x}$ gives

$$
\frac{\partial L}{\partial \boldsymbol{x}} = \sum_i \sum_j \frac{\partial w^i}{\partial \boldsymbol{x}} w^j \boldsymbol{W}(\boldsymbol{A}^i \boldsymbol{x} + \boldsymbol{b}^i)(\boldsymbol{W}\boldsymbol{A}^j \boldsymbol{x} + \boldsymbol{W}\boldsymbol{b}^j - \boldsymbol{y}) + w_i w_j \boldsymbol{A}^{i\top} \boldsymbol{W}^\top (\boldsymbol{W}\boldsymbol{A}^j \boldsymbol{x} + \boldsymbol{W}\boldsymbol{b}^j - \boldsymbol{y}),
\tag{14}
$$

where the right term reduces to a convex combination of the gradient of the linearized physics based forward models, modulated by some matrix $\boldsymbol{W}$, when $i = j$.

## D   Extensions to Interpolation: Additional Experiments

In this section, we begin a preliminary study on PETAL for interpolation (i.e. a random split of the available data). By removing the dynamics shift between the training and the validation set, we are able to explore more complex models beyond linear components.

### D.1   Selecting Reference Points

Due to the random split, we can no longer rely on heuristics to select appropriate reference models. Instead, we perform a projected $k$-means approach to select our reference points. The standard

$k$-means algorithm is applied to all SSPs in the training set. Once the algorithm terminates, the closest point in the training set to each mean is selected as the reference point to be embedded in the model. For our experiments, we used $k = 20$.

## D.2  Extending the layers

We maintain the overall structure of PETAL by retaining an SSP encoder-decoder, an embedded physics module, a learned weighted average module, and a final layer for the predicted arrival time. However, we extend the learned components by replacing the single linear layer used in the main manuscript with MLPs.

In addition to replacing the final linear layer $W$ with a series of linear layers followed by a leaky ReLU activation, we also include a skip connection between the output of the learned weighted average module and the final arrival times prediction. We also observed that including an auxiliary loss in the form of mean squared error between the ground truth arrival times and the direct output of the weighted average layer (prior to passing through the final layers) was beneficial.

We experimented with replacing linear layers with convolutional layers, but noticed a huge degradation in performance. We hypothesize that this is due to the nature of the data not being suitable for convolutions. The input SSPs are not translationally invariant depth-wise. The output arrival times are spatially small ($20 \times 20$), and thus the choice of padding can easily introduce errors at the edge, degrading the final prediction.

However, we did notice introducing convolutional layers to have a beneficial effect when added to the end of the SSP decoder. The convolutional filters play the role of filtering/smoothing the predicted output to create more realistic reconstructions.

## D.3  Results

We compare all our results against a Gaussian process regression (GPR) baseline. We construct a dataset by randomly sampling 1000 SSPs from the 1439 available from each of the 10 slices, totaling 10,000 total training samples. We use a random subset of 1000 points to train GPR due to memory limitations of the method. Two models (MLP and PETAL) are trained to approximate the forward model and then used in an optimization framework for inference.

The MLP consists of 4 hidden layers with dimension 1500 for a total of 5 linear layers separated by leaky ReLU. Batch normalization was found to degrade performance and thus was omitted for the final model.

The final PETAL architecture uses a 2 layer MLP with hidden dimension 1000 for the $x$ encoder and $x$ decoder separated by leaky ReLUs. The $x$ decoder also uses 3 convolutional layers with kernel size $3 \times 7$ and 32 filters. Finally, the output layer consists of a 3 layer MLP with hidden dimension 800.

The performance of each model is shown in Table 6. We experiment with gradient descent (SGD) and limited memory Broyden–Fletcher–Goldfarb–Shanno algorithm (LBFGS)[18] as the optimization algorithm. For both MLP and PETAL, LBFGS reached a local minima significantly faster than SGD. However, it did not always achieve the most accurate reconstruction, likely getting trapped in local minimas due to the non-convex nature of optimizing a neural net.

MLP achieved higher accuracy for emulating the forward model. However, this higher performance did not translate to the inference stage. MLP was able to perform well when given a more accurate starting point (GPR init), but failed to outperform GPR with the average intialization, getting caught in local minimas. PETAL was able to outperform GPR regardless of initialization. This suggests that the structure embedded into PETAL's architecture aids in the optimization process when determining descent directions. Examples of reconstructed SSPs can be found in Figure 8. Note the artifacts and hallucinations introduced by optimizing the MLP surrogate.

# E  Limitations

Our proposed model was evaluated on noise-less simulations, both with respect to measurements and sensor/receiver placement, which is not true in practice for data collected in the real world. We also did not explore the selection process of the reference points to linearize around, assuming that the

Table 6: RMSE (m/s) of random data split

| Model | Optimization | Avg Init | GPR Init |
|-------|-------------|----------|----------|
| GPR | — | 0.176 | — |
| PETAL | SGD | 0.144 | **0.128** |
| PETAL | LBFGS | 0.142 | 0.129 |
| MLP | SGD | 0.224 | 0.146 |
| MLP | LBFGS | 0.231 | 0.164 |

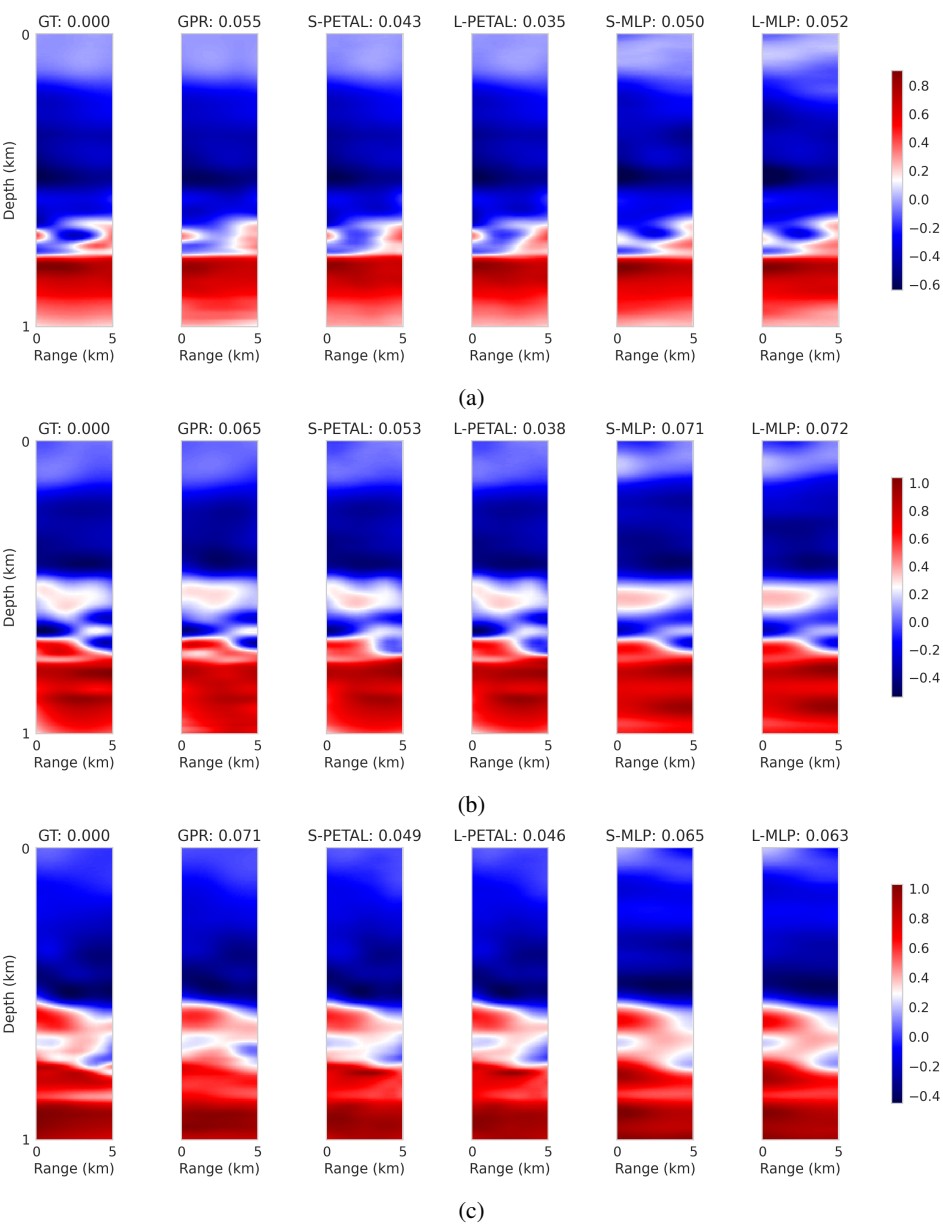

Figure 8: Example visualizations of predicted sound speed profiles for each method. The "S-" prefix denotes models optimized with SGD and "L-" with LBFGS.

chosen subset sufficiently represented the data. However, section B.2 suggests that the selection of reference points is somewhat robust to unseen dynamics.

