# OpenReview forum: "PETAL: Physics Emulation Through Averaged Linearizations for Solving Inverse Problems"
_NeurIPS.cc/2023/Conference — NeurIPS 2023 poster_

### Official Review · Reviewer_Bbps · 2023-06-25

**Soundness:** 3 good
**Presentation:** 3 good
**Contribution:** 2 fair
**Rating:** 6
**Confidence:** 3

**Summary:**

This paper tackles the area of inverse problems using a neural network architecture. Specifically this follows the neural adjoint paradigm, whereby the algorithm learns a forward model and then uses it for inference. The authors expand on this existing methodology by incorporating knowledge of the physics-based forward model and using a latent space in the optimization stage.

**Strengths:**

The paper is clearly written and well motivated, providing background and alternative methodologies. The paper also has clear mathematical notation and is benchmarked on a simulated system that appears grounded in reality, i.e. the sounds speeds in the Gulf of Mexico. The authors also attempt to tease out and explain the effects of the various architecture components using ablation studies.

**Weaknesses:**

The paper doesn’t discuss the hyper-parameters of the networks used and their effects on the results. When discussing the performance of their algorithm against the MLP, it is unclear what this MLP actually is and while the MLP doesn’t appear to perform as well, it is rather a close second. Hence, the extra complexity of the PETAL algorithm and the benefit of the embedded physics is not as transparent as it is stated in the paper.

**Questions:**

1. Please include more details regarding the hyper-parameters of your PETAL network.
2. Please include details about the MLP used in the benchmarking.
3. What would the performance of the MLP be if the embedded physics is also provided to it?
4. I am not convinced based on Table 1 that PETAL is more robust to initialization compared to the MLP, as it is claimed on page 8 line 281. It could also be chance, given how small the differences are between the various initializations for both PETAL and the MLP. Please provide further evidence for this claim.
5. How many simulations of the Gulf of Mexico have been done? If only one, I would suggest further benchmarking on either other simulations of this Gulf or even more beneficial, on other inverse problems.

**Limitations:**

Yes

---

> ### Author Rebuttal · Authors · 2023-08-07
>
> We thank the reviewer for their thoughtful feedback. In particular, we are glad the reviewer found our work clearly written and well motivated. We attempt to address some of their issues below:
>
> > Hyper-parameters of PETAL and MLP
>
> We include the sizes of the various layers included in PETAL in Appendix A.2 as well as training hyperparameters. The details of the MLP are included in A.3. We are happy to include any additional detail suggested by the reviewer.
>
> > What would the performance of the MLP be if the embedded physics is also provided to it?
>
> The suggestion of providing embedded physics to an MLP model is a interesting. If we consider the case of stacking the output across all the reference linearized models and feeding this into an MLP, I would expect the performance to be potentially worse than PETAL. The primary reason is that the weights learned to merge the different outputs would be fixed, while as PETAL produces adaptive weights depending on the proximity of the input to each specific reference point. Fixed weights might be more accurate if each of the models had some general approximation error, but that is not the case. These linearizations are more accurate within a small neighborhood of the reference point used and thus their accuracy should vary depending on the input.
>
> We can also consider the case of using PETAL’s learned adaptive weighting scheme to combine the outputs of the embedded physics module before feeding it into an MLP, but I would argue that this is simply an extension of the proposed model with a non-linear final output layer instead of the current linear one.
>
> > I am not convinced based on Table 1 that PETAL is more robust to initialization compared to the MLP, as it is claimed on page 8 line 281. It could also be chance, given how small the differences are between the various initializations for both PETAL and the MLP. Please provide further evidence for this claim.
>
> We have extended the experiments, and included additional tables of results. Reviewing Table 1, we can see that MLP lead to degradation in accuracy, despite being initialized with the LFM/Tik estimate which is more accurate than the training set average (approximately 1 m/s RMSE), suggesting that it gets caught in a local minima. Alternatively, PETAL is able to refine the answer further across different realizations (different trained weight initialization, but fixed initialization for optimization for inference purposes).
>
> Further examination of Table 3 also shows similar trends. PETAL is able to achieve better results when given a better starting point in all three variability scenarios for the dense $20\times20$ forward model and for the low variability scenario for the spare $10\times10$ configuration. However, it does fail to achieve as good of performance for the medium and high variability scenario. This is likely due to the more challenge forward model setting with less observations making the optimization problem more difficult to solve. Alternatively, MLP shows more sensitivity to its input, only succesfully achieving better results with a better initialization in the low variability regime.
>
> > How many simulations of the Gulf of Mexico have been done? If only one, I would suggest further benchmarking on either other simulations of this Gulf or even more beneficial, on other inverse problems.
>
> While we have not finished designing experiments for other domains, we have performed additional experiments on different Ocean Acoustic examples. We experimented with a sparse source/receiver pair configuration ($10\times10$ instead of $20\times20$) as well as on a different portion of the ocean for the full $20\times20$ configuration. All models followed similar trends (i.e. PETAL outperformed MLP which outperformed linearizations). Please see the attachment for summarized results.

---

> > ### Comment · Reviewer_Bbps · 2023-08-10
> >
> > Dear authors,
> >
> > Thank you for the time taken to write the rebuttal and addressing the points I raised.

---

### Official Review · Reviewer_1TNX · 2023-07-04

**Soundness:** 2 fair
**Presentation:** 2 fair
**Contribution:** 1 poor
**Rating:** 3
**Confidence:** 3

**Summary:**

The paper proposes a new neural network architecture to simulate the evolution of a given physical system that can be seen as a generalized weighted combination of linearizations around a set reference pairs of state and measurements $(x_{ref, i}, y_{ref, i})_{i=1}^{N}$. The paper proposes to use this method when solving inverse problems in physics through MAP (maximum a posteriori) estimations. It then shows that using this architecture they can achieve smaller RMSE w.r.t the state $x$ that was used to generate the observation $y$ with different regularizations (priors) for the problem of ocean acoustic tomography when compared to other forward models.

**Strengths:**

The paper proposes a reasonable model that is based on reference set. The presentation of the inverse problem applications to the physics domain is good and the application to ocean acoustic tomography is original and not often found amongst the NeurIPS community.

**Weaknesses:**

One of the main weakness of the paper is the lack of support of the claim that the proposed architecture is a efficient tool for modelling general forward problems in physics. There are two main problems with the validation of the method:

1. Lack of replicates for the validation metrics:
 - Neither table 1 or 2 contain confidence intervals. Those kind of metrics are expected to have aggregated results from several replications of initializations of the network before training as well as several initializations of the optimization problem for solving the inverse problem. Without it, it is quite hard to know if the RMSE differences are signficative in say MLP vs Petal in table 1 or PETAL vs WAN in table 2.

2. Comparison with other baselines:
I think the comparison with other possible forward models is lacking. Indeed I'd expect a comparison to a more expressive meta model than a MLP, by either training a more complex network that better exploits the 2d nature of the problem. Other methods that take advantage of some reference set, such as Gaussian processes could have been exploited as well.
I'd like to know as well if the proposed method scales to other domains than Ocean Acoustic Tomography. This would be of great interest in reproducibility, since the dataset used is not available and therefore the community has no way of replicating the experiments.


**Questions:**

* If $P_y$ is a matrix, then I do not see why the choice of separating $W$ and $P_y$ in equation 8, since $WP_Y$ is just another matrix of dimension $2431 \times 800$, that could be directly parameterized.

* It is known that self-normalisation of weights that correspond to inner products in big dimensions lead to weight collapse (after self normalisation one of the reference ($x_ref, y_ref$) can have a weight of almost one while the others 0). In this case, the inverse problem would be basically based upon $A_ref$ for this given pair. Is this something that was seen during the optimization for solving the inverse problem? If this is the case, the model would be basically selecting one of the reference solutions, and could be simplified.

* How sensitive is the model to the reference solution that were chosen? How would it be impacted if another set was chosen?

*The MLP model seems to be overfitting (lines 275 and 276). If i'm not mistaken, standard techniques to reduce overfitting have not been tried (dropout, batch normalisation, etc...). How would those impact the performance?

---

> ### Author Rebuttal · Authors · 2023-08-07
>
> We thank the reviewer for their thorough review. We respond to their concerns below:
> > Neither table 1 or 2 contain confidence intervals.
>
> We repeated the model training as the reviewer suggested and have updated Table 1 in the author rebuttals attachment to reflect the confidence intervals. The reviewers suggestion was helpful in further understanding our model. We observe that PETAL seem more sensitive to model initialization than MLP with its higher standard deviation, but still manages to outperform other baselines.
>
> > Comparison with other baselines. Indeed I'd expect a comparison to a more expressive meta model than a MLP, by either training a more complex network that better exploits the 2d nature of the problem.
>
> We had similar thoughts to the reviewer for data-driven surrogate models with respect to exploiting the 2d nature of the problem. Beyond simple MLPs, we also tested fully convolutional models, both 1d and 2d, as well as hybrid convolutional/linear models. We also experimented with encoder-decoder structures, mixing and matching linear/convolutional/deconvolutional layers. Unfortunately, all models underperformed during the first stage of attempting to learn the forward model with the exception of MLP and PETAL. We conjecture that this is partially due to the changing statistics between training and validation set, making it more difficult to train a generalizable model. We did an experiment with a random split of train/validation/test, and all models achieved very good performance. However, this does not reflect a realistic data split, hence the current split used in the manuscript.
>
> > Other methods that take advantage of some reference set, such as Gaussian processes could have been exploited as well.
>
> We thank the reviewer for bringing this to our attention. We will make sure to thoroughly investigate this alternative method for future work.
>
> > I'd like to know as well if the proposed method scales to other domains than Ocean Acoustic Tomography. This would be of great interest in reproducibility, since the dataset used is not available and therefore the community has no way of replicating the experiments.
>
> We agree with the reviewer that empirically showing the proposed model on a variety of forward models would best highlight its potential to solve inverse problems. We are actively investigating other potential models and datasets, but do not have available results to share just yet. However, we have performed additional experiments on the simulation of the Gulf of Mexico as shared in the overall rebuttal. More specifically, we repeat a subset of experiments on a different region of the ocean and we simulate a different forward model configuration on the region presented in the manuscript. Both new experiments match the trends of the original experiment with our proposed model performing the best.
>
> > If $P_y$ is a matrix, then I do not see why the choice of separating...
>
> The reviewer is correct. Notation was done to reflect architecture design choices in the code related to attention layers, but are otherwise indeed redundant. We will simplify the notation to a single matrix in the final revision.
>
> > It is known that self-normalisation of weights that correspond to inner products in big dimensions lead to weight collapse (after self normalisation one of the reference ($x_{ref}$, $y_{ref}$) can have a weight of almost one while the others 0). In this case, the inverse problem would be basically based upon $A_{ref}$ for this given pair. Is this something that was seen during the optimization for solving the inverse problem? If this is the case, the model would be basically selecting one of the reference solutions, and could be simplified.
>
> The reviewer makes an insightful observation. Indeed, we found similar results in our initial experiments where the predicted weights collapsed into favoring a single $A_{ref}$ across a number of examples. However, we addressed this by applying spectral normalization to the projection matrix. This led to outputs being at an order of magnitude that did not collapse to a delta after the soft-max function. Instead we observe a weight distribution that changes depending on the input.
>
> > How sensitive is the model to the reference solution that were chosen? How would it be impacted if another set was chosen?
>
> For the scope of this manuscript, we did not experiment extensively with the selection of reference models, instead relying on a heuristic (the most recently available point in the trainset). However, we did conduct one experiment in the appendix (B.2) where we construct and train the model with references only from 9 slices before testing it on the 10th slice. There was very little degradation in performance, suggesting (at least in this instance) some robustness to the reference set. We are currently exploring a k-means clustering approach to selecting the reference model but are not ready to comment at this point.
>
> > The MLP model seems to be overfitting (lines 275 and 276). If i'm not mistaken, standard techniques to reduce overfitting have not been tried (dropout, batch normalisation, etc...). How would those impact the performance?
>
> During experiments, we did test various methods to reduce overfitting including batch normalization and dropout. However, this led to a degradation in performance on the validation set, so we chose to not include them in the final model. We note that some of the issues arise from the nature of our dataset and how we chose to split it (sequentially in time).

---

> > ### Comment · Reviewer_1TNX · 2023-08-12
> >
> > I thank the authors for addressing all the points I raised, but I still feel that in the current format there is still lacking evidence that the proposed model is indeed a serious candidate for a good architecture solving inverse problems in more general physics problems. I'd expect it to be tested in different scenarios with different physical problems and with more competitive architectures than MLP. I see that other architectures do not seem to work better than MLP in this example, which I find puzzling and for me makes the need of another example even bigger.

---

> > > ### Author Response · Authors · 2023-08-15
> > >
> > > We acknowledge that testing the proposed model on additional set ups is worth doing, and is something we're actively pursuing.
> > >
> > > However, as a discussion point, perhaps one of the reasons why "more competitive architectures" fail to work well in our case may be due to the nature of the data itself. Although it's shaped similar to images ($N_c\times H\times W$ where $N_c=$ num channels), it doesn't obey the same properties as images. For example, the input is a range by depth ($11\times231$) regularly discretized grid with different resolution in each direction. It has some sense of spatial invariance in the range dimension, but not in depth making learned convolutional filters a questionable choice (deeper parts of the ocean have different properties compared to the surface).
> > >
> > > For the output, although it is organized in a $2\times20\times20$ tensor, it doesn't have the same spatial relationship between "pixels" as an image normally would. The $20\times20$ corresponds to all the different source-receiver pairs, but a relationship between one set of $3\times3$ subset would not necessarily hold for another subset, rendering the translational invariance property of convolutional filters invalid. We hypothesize that MLPs were able to perform reasonably well on this task (as well as other related works) because it is more flexible for non-image data.
> > >
> > > These models trained on image datasets are indeed powerful tools, but they don't always translate to scientific data. This is a huge motivional factor for us seeking out alternative architectures to handle this kind of data and what lead to us proposing the PETAL architecture detailed in the manuscript.

---

### Official Review · Reviewer_yV52 · 2023-07-05

**Soundness:** 3 good
**Presentation:** 2 fair
**Contribution:** 2 fair
**Rating:** 6
**Confidence:** 3

**Summary:**

A surrogate model based on multiple linearizations of the target simulator is proposed. It is then used for solving inverse problems by gradient descent.

**Strengths:**

- The idea of using linearizations for surrogate model is interesting.
- The explanation about the better gradients of such a model is reasonable.
- The method is examined with a concrete example.
- The rationale of why the proposed model's gradient can be better is very nice.

**Weaknesses:**

- The difference from previous studies [4,10,30] is unclear, which is the main reason why I lean to rejection currently.
- The experiment is conducted only for a single forward model.

**Questions:**

1. What is common or different from the previous studies [4, 10, 30]? The authors mention that their work departs from them, but a concrete explanation of the relation is missing.

2. In the proposed model, how do you choose $x_ \text{ref}$ in general?

3. What is the role of $W$ in Eq. (3)? Isn't this redundant with $P_y$?

----

Post-rebuttal: The questions above are basically solved after the rebuttal.

---

> ### Author Rebuttal · Authors · 2023-08-07
>
> Thank you for your thoughtful review. We respond to the highlighted questions below:
>
> > What is common or different from the previous studies [4, 10, 30]? The authors mention that their work departs from them, but a concrete explanation of the relation is missing.
>
> - [4] ML Emulation of gravity wave
> This work primarily focuses on building surrogate models. Although they include the surrogate as part of a pipeline that includes physics-based components, the learned model only serves to replace a specific stage in a data-driven manner. Our proposed methodology is similar in that it is attempting to approximate a physics-based process (“forward model”), but differs in that it incorporates low-fidelity versions (linerizations at different reference points) of the forward model directly into the surrogate, so it is not a purely data-driven method.
> - [10] Tangent-linear and adjoint models
> This work proposes a surrogate model in the form of a multi-layer perceptron net (MLP) and discusses tests to verify the accuracy of both its output as well as the gradient for inversion. They construct/test the tangent-linear (linearization of the forward model around a reference point) as well as the adjoint (linearization applied for gradient calculation), but **for the learned surrogate model**. We use an ensemble of tangent-linear models (referred to as jacobians in the manuscript) of the **true physics-based forward model** at a number of reference points rather than the learned surrogate when constructing our architecture. We also compute the adjoint of the surrogate when performing inversion, but do so implicitly through the use of auto-grad libraries and note that this specific approach is not our primary contribution, but that of [30].
> - [30] Benchmarking Inverse Problems and the Neural Adjoint Method
> While our work is an extension of [30], we believe that tackling much larger dimensional problems (around dimension 10 vs 1000) and providing the tools to do so in the form of a learned basis for improved reconstruction is a notable contribution. This also departs from other previous work such as [13] that used principal component analysis to generate a basis.
> Furthermore, our primary contribution lies in a suggestion for a model architecture that takes advantage of a specific type of low-fidelity physics model (linearizations around references). [30] describes a methodology assuming a generic black-box architecture that is not necessarily tailored to solving inverse problems. Our proposed model can even be the foundation for more complex surrogate models (additional layers to further refine the output).
>
> > In the proposed model, how do you choose $x_{ref}$ in general?
>
> Currently we are employing a heuristic approach based on current practices (i.e. selecting the most recently available point for linearization). We are currently exploring a k-means clustering approach to select a reference set for better coverage, but are currently not ready to comment on such an approach.
>
> > What is the role of $W$ in Eq. (3)? Isn't this redundant with $P_y$?
>
> The reviewer is correct. Notation was done to reflect architecture design choices in the code related to attention layers, but are otherwise indeed redundant. We will simplify the notation to a single matrix in the final revision.
>
> > The experiment is conducted only for a single forward model.
>
> We agree with the reviewer that empirically showing the proposed model on a variety of forward models would best highlight its potential to solve inverse problems. We are actively investigating other potential models and datasets, but do not have available results to share just yet. However, we have performed additional experiments on the simulation of the Gulf of Mexico as shared in the overall rebuttal. More specifically, we repeat a subset of experiments on a different region of the ocean and we simulate a different forward model configuration on the region presented in the manuscript. Both new experiments match the trends of the original experiment with our proposed model performing the best.

---

> > ### Comment · Reviewer_yV52 · 2023-08-13
> >
> > Thanks for the response, it helped a lot to understand the context. Accordingly I increased the score (4 --> 6). I would suggest adding (a part of) the clarification of the relation to previous studies somewhere in the paper.

---

### Official Review · Reviewer_exJb · 2023-07-09

**Soundness:** 2 fair
**Presentation:** 4 excellent
**Contribution:** 3 good
**Rating:** 6
**Confidence:** 4

**Summary:**

This paper further improves a method used in reference [30]. The goal is to reconstruct the inverse problem, through a "gray-box" model. For the forward solver, a parametrized model is proposed: many linearizations centered at different (reference) points of the nonlinear forward problem are ensembled. The ensemble weights depend on a learned similarity measure. The inverse problem is solved using the usual way (iterative methods with a regularization) with no parameterization involved.

Overall, I think this is a decent submission and lean towards acceptance.


**Strengths:**

- This paper is clear-written and well-motivated.
- There is a full-blown systematic ablation study.

**Weaknesses:**

- The contribution versus that of [30] seems limited.
- Certain presentations are somewhat focused on a specific inverse problem (ocean tomography), for example, what is a "ray tracer" on line 134? What is an "eigenray"? What precisely is the forward model in this case?
- The biggest weakness is the assumption that the Jacobian of the forward operator can be straightforwardly obtained in equation (6). Using EIT as an example, how to measure the change in the nonlinear forward operator with respect to the conductivity coefficients is still quite unsolved.
- The error for fitting the forward problem is not reported.
- The reason why, or an empircal study of, not updating the parameters of the forward problem solver during the iterative procedure to solve the inverse problem is not mentioned, at all.
- Missing a whole line of reference on EnKF for inverse problems.
- The baseline for comparison is not that strong.


There are some grammatical errors:
- line 23: "well known" -> "well-known".
- line 53: there should be an article before "simulation".
- line 93: "physics obeying" -> "physics-obeying".
- line 104: "physics based" -> "physics-based" and a few other places as well.
- line 105: a minor nitpicking, "Pytorch" should be "PyTorch".
- line 154: "dot-product" should be "dot product" here, different from that in "dot-product attention".
- line 173: "undesireable local minimas" should be "undesirable local minima", same problem on line 279.
- line 190: "trade offs" -> "trade-offs".
- line 193: "generate" -> "generates".
- line 214: "is essential" -> "are essential".
- line 224: "a SSP" -> "an SSP".
- line 232: "range dependent" -> "range-dependent".
- line 281 and 312: "out-perform" should be one word without the hyphen.
- line 314: "each contribute to ..." -> "each contributes to".


**Questions:**

- line 68: while the meaning of "TV" is obvious in the context, it is better to define it before using the abbreviation for the first time.
- What does it mean that "direct inversion methods" yield a "single estimate"?
- line 133: $n$ the total number of "observations"?
- line 162: What is the spectral "norm", based on the context right before it should be "normalization", not a norm, please confirm this.
- What does "embedding physics" in Section 3 mean?

**Limitations:**

The authors did not discuss any limitations.

---

> ### Author Rebuttal · Authors · 2023-08-07
>
> Thank you for your thorough and thoughtful review. We are delighted that you found our work well-motivated. We respond to your highlighted issues below:
>
> >The contribution versus that of [30] seems limited.
>
> While our work is an extension of [30], we believe that tackling much larger dimensional problems (around dimension 10 vs 1000) and providing the tools to do so in the form of a learned basis is a notable contribution. This also departs from other previous work such as [13] that used principal component analysis to generate a basis.
> Furthermore, our primary contribution lies in a suggestion for a model architecture that takes advantage of a specific type of low-fidelity physics model (linearizations around references). [30] describes a methodology assuming a generic black-box architecture that is not necessarily tailored to solving inverse problems. The proposed model can even be the foundation for more complex surrogate models (additional layers to further refine the output).
>
> > Certain presentations are somewhat focused on a specific inverse problem (ocean tomography), for example, what is a "ray tracer" on line 134? What is an "eigenray"? What precisely is the forward model in this case?
>
> To expand on section 4 (line 212), the forward model simulates emitting a sound wave from each of the 20 sources. The receivers aim to detect peaks of the observed pressure wave, corresponding to the "arrival time" of the direct path and the surface bounce path of the sound wave, leading to a toal of 800 observables. The simulation is an approximation of the wave equation for the pressure waves. Rather than solving the full partial differential equations, we model the wave as an eigen-ray traversing through space, reducing it to an ordinary differential equation solve along the path of the ray between the source and receiver.
>
> The ray tracer refers to BELLHOP, a program used in the ocean acoustics community. This program is available freely at the Ocean Acoustics Library (http://oalib.hlsresearch.com/AcousticsToolbox/). We will add this to the final revision for clarity.
>
> > The biggest weakness is the assumption that the Jacobian of the forward operator can be straightforwardly obtained in equation (6). Using EIT as an example, how to measure the change in the nonlinear forward operator with respect to the conductivity coefficients is still quite unsolved.
>
> The reviewer brings up a very important critique. Although we use explicit Jacobians in our model formulation, it is not a requirement. The methodology outlined in our paper can also take advantage of implicit formulations, since we only need the ability to apply the Jacobian to arbitrary vectors, as is the case in any problem that uses the adjoint state method (computing Jacobian-vector products).
>
> We would also like to point out that the full Jacobian can be calculated using the same adjoint calculation by applying different vectors (e.g. elementary). Although this is expensive and would scale with the dimension of the output, it can be treated as an offline cost and would only need to be done for a small subset (the reference points selected). However, we do acknowledge that the need to apply the Jacobian in some form (explicit or implicit) does somewhat limit our method to a certain subclass of problems.
>
> > The error for fitting the forward problem is not reported.
>
> PETAL achieves a (normalized by training set statistics) RMSE of 4.88e-2 and MLP 3.62e-2. The (normalized) validation RMSEs are 1.02e-1 and 1.22e-1, respectively. We report a validation (unnormalized) rmse of 4.98e-4 seconds for arrival times for PETAL and 6.08e-4 seconds for MLP in the appendix (A.2 and A.3). We will revise the final draft to include the training numbers.
>
> > The reason why, or an empircal study of, not updating the parameters of the forward problem solver during the iterative procedure to solve the inverse problem is not mentioned, at all.
>
> We followed the techniques practiced in the underwater acoustic community (a single linearization used across a range of samples) for computational reasons. We did a preliminary study on re-linearizing at intermediate points (manuscript submitted but not available for release yet) and found that on the full dense setup, it lead to minimal improvement. However, it did improve the estimation results further on sparser source/receiver pair set ups ($10\times10$ and $3\times10$).
>
> > Missing a whole line of reference on EnKF for inverse problems.
>
> We thank the reviewer for bringing this to our attention. We will make sure to include the relevant references in the final revision.
>
> > Grammatical errors/suggestions:
>
> We will edit these in the final revision. Thank you for pointing these out.
>
> > What does it mean that "direct inversion methods" yield a "single estimate"?
>
> Non-bayesian models that focus on learning a mapping from $y$ to $x$ only produce a single estimate of $x$ for each observation $y$. This contrasts from optimization based methods that can produce different answers with different initializations
>
> > Spectral "norm"
>
> The reviewer is correct. We have updated the manuscript to reflect that it is a "normalization".
>
> > What does "embedding physics" in Section 3 mean?
>
> We use embedding physics in this context to mean directly incorporating physics-derived (low-fidelity approximation) models into the machine learning architecture.

---

> > ### Comment · Reviewer_exJb · 2023-08-10
> >
> > I, Reviewer exJb, have read the rebuttal from the authors (as well as the ones to other reviewers). The paper is definitely one of the betters ones among the batch I reviewed.
> >
> > However, I do want to raise a question to the AC about the data availability: the difficulty to get the data definitely justifies it being proprietary, yet does this need to be flagged for ethics review?
> >
> > To address this point, it would be nice if the authors can add the more detailed tensor shapes of the input, and the latent representations propagating in the hidden layers, instead of "input dim" and "output dim". For example, like the ones used in PyTorch docs (see Parameters section of https://pytorch.org/docs/stable/generated/torch.nn.MultiheadAttention.html).

---

> > > ### Author Response · Authors · 2023-08-11
> > >
> > > Thank you for the response. We're pleased that you find our work interesting!
> > >
> > > Please see the more detailed description of the models below:
> > > Let BS = batch size
> > > PETAL:
> > > - x encoder (Linear)
> > >   - input ( BS, N_x) where N_x = 2541 (size of ssp vectorized)
> > >   - output (BS, H_x) where H_x = 1000
> > > - Embedded Physics Layer (ensembled linear forward models)
> > >   - input (BS, N_x) where N_x = 2541
> > >   - output (BS, N_ref, N_y) where N_y = 800 (number of observations), N_ref = number of reference linearizations
> > > - Weighted Average Layer (Attention Layer)
> > >   - Query - (BS, 1, H_x), where E_x = 1000 is the learned latent size of the x Encoder
> > >   - Key - (BS, N_ref, H_x), where N_ref = number of reference linearizations
> > >   - Value - (BS, N_ref, N_y) where N_y = 800
> > >   - output - (BS, 1, H_out) where H_out = 1000
> > > - x decoder (Linear)
> > >   - output (BS, H_in) where H_in = 1000
> > >   - input ( BS, N_x) where H_out = 2541 (size of ssp vectorized)
> > > - y decoder (Linear)
> > >   - output (BS, H_in) where H_in = 1000
> > >   - input ( BS, N_y) where H_out = 800 (number of observations)
> > >
> > > We are currently refactoring the Weighted Average Layer to simplify it. However, the current implementation is an extension of PyTorch's attention layer so it applies a projection on the query/key (extended to be shared) that maps (BS, *, H_x) -> (BS, *, E) where E = 1000 is the embedding size of the attention layer. Likewise, a similar projection is learned for the value that maps (BS, N_ref, H_y) -> (BS, N_ref, E). Once the values are summed up, it passes through a final Dotprod Out layer that maps (BS, 1, E) -> (BS, 1, H_out). See Figure 2 of https://arxiv.org/pdf/1706.03762.pdf for more details.
> > >
> > > The MLP consists of a series of linear layers followed by leaky ReLUs.
> > > - Input layer
> > >   - input (BS, N_x) where N_x = 2541
> > >   - output (BS, H) where H = 1500
> > > - Hidden layers (x4)
> > >   - input (BS, H)
> > >   - output (BS, H)
> > > - Output layer
> > >   - input (BS, H)
> > >   - output (BS, N_y) where N_y = 800
> > >
> > > Please let us know if there's any other details you'd like us to share!

---

> > > > ### Comment · Reviewer_exJb · 2023-08-11
> > > >
> > > > Thanks for the response. I have raised my score from 5 to 6.

---

### Author Rebuttal · Authors · 2023-08-07

We thank all the reviewers for their time and thoughtful feedback. We have included an additional set of experiments as well as a more detailed breakdown of the scenario presented in the manuscript.

For Table 1, we retrained all neural net models 10 different initialized weights before performing the optimization for inference with fixed input initializations. We have included the standard deviation for MLP and PETAL to showcase the confidence intervals. Note that the proposed models upper end still falls below that of other models.

For Table 2, we perform initial experiments on a different region of the Gulf of Mexico simulation. We still use 10 slices with the same train/validation/test split.

For Table 3, we divide the original dataset into three scenarios based on the variability seen in the test set (last 239 samples) for each slice. Slices are sorted based on the average root mean square error (RMSE) from the average of the first 1000 samples (corresponding to the train set) for each respective slice. This approximates how much the slice deviates over time. The low variability slices had around 1.6 m/s deviation, the medium 2.5 m/s, and the high 3.4 m/s.

We then repeat the experiments for a different configuration of the forward model. Instead of using $20\times20$ source/receiver pairs, we reduce this to a $10\times10$ configuration, leading to signficantly less observations. We would expect performance to degrade in this sparser scenario, which is reflected in the experimental results. However, general trends still hold with the trained surrogates out-performing the classical baselines and the proposed PETAL model performing the best.

One interesting thing to observe is that although there is some degradation in performance as variability increases, this does not always hold true. For example, PETAL actually performs better in the higher variability scenario. We hypothesize that this is due to being trained on all 10 slices rather than each individual slice and thus learning a more generalized forward model.

---

### Decision · Program_Chairs · 2023-09-21

**Decision:**

Accept (poster)

**Comment:**

This paper describes a new method for solving inverse problems that rely on complex physics-based forward operators. The paper proposes to train a differentiable surrogate forward emulator using an interesting learned model that uses linearizations around reference points.
The authors did a great job in their rebuttal by including additional results for fewer source/receiver pairs in a much more under-sampled regime. The fewer observations lead to graceful degradation as the additional experiments show.
One reviewer remains concerned about limited evaluation to more general physics problem but overall I think there is sufficient novelty in this paper.